# Personalized Federated Learning for Medical Segmentation using Hypernetworks

**Hilit Segev & Gal Chechik**
Bar-Ilan University

## Abstract

In federated learning (FL), several clients jointly train a shared model without sharing their data, maintaining data privacy and reducing communication costs. In personalized federated learning (PFL), each client has their own model, and models are trained jointly. Hypernetworks have been shown to be useful for PFL in classification problems, but it is still not clear how to apply them to problems like segmentation. There, models are very large, and it is not known what parts of models should be personalized, and what parts should be shared across clients. Here, we explore HNs for PFL for solving a problem of image segmentation in the context of medical imaging diagnosis. Using MRI scans for prostate segmentation, we demonstrate that using a hypernetwork to personalize a single convolution layer and the batch-norm layer outperforms local and FL baselines.

## 1 Introduction

Federated Learning (FL) (McMahan et al., 2017) is the task of learning a single model across different datasets, each held by a different client. FL is most valuable when clients cannot share their data due to concerns about privacy, communication, or storage. This is the case for instance, in segmentation of medical imagery, where different hospital are forbidden from sharing patient data with other institutions. This greatly limits the volume and richness of training data, reducing model quality and robustness. Training shared models without sharing data holds great promise in this area (Roth et al., 2021; Wang et al., 2022). Often, data of different clients comes from different distributions, due to different sensors or annotation procedures. In such cases, training one shared model for all clients may perform poorly. Personalized Federated Learning (PFL) Kulkarni et al. (2020); Dinh et al. (2020); Achituve et al. (2021) aims to solve this problem by learning a model for each client while benefiting from a joint training procedure.

Hypernetworks (HNs) (Ha et al., 2017), have been shown to provide an effective solution to PFL problems (Shamsian et al., 2021; Lin et al., 2023; Yang et al., 2022). HNs are deep networks that emit the weights of other deep network models. In the context of PFL, HNs have been trained for small-scale classification problems, but it is not known if and how they can be extended to segmentation where models are much larger. HNs were used for segmentation, but not in the context of PFL (Yuval Nirkin, 2021). There the HN was conditioned on local image regions, and produced weights of a decoder network per patch.

We describe the first HN-based PFL for segmentation of medical images. Specifically, we develop a HN for segmenting prostates in 3D MRI scans, that can learn from different clients in a FL manner. Applying HNs to segmentation poses new challenges since segmentation models tend to be very large, and it is not clear how and what parts of these models should be generated by the HN.

## 2 Our approach

In our setup, a central server runs a HN. The HN receives as input a descriptor of a client, and emits weights of a target segmentation model for that client. Specifically, each target network is a 3D-UNet model (Çiçek et al., 2016). The HN predicts the weights of its last convolution layer, which we call *personalized weights*. The remaining target weights – *global weights* – are shared across all clients and are learned at the server using a federated approach, specifically, FedSGD (McMahan et al., 2017). We name this approach **HyperSeg Conv**.

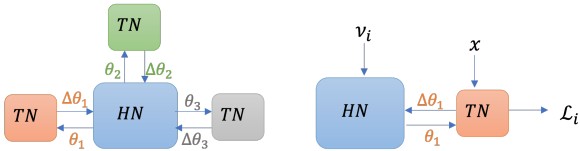

Figure 1: **Left:** A hypernetwork with 3 target networks. In our case, the HN is located at a central server and each target network is located in a separate client. **Right:** the input to the HN is a descriptor $v_i$ of the target network $i$, for client $i$. See appendix E for the full architecture.

We further tested two extensions of this approach. First, we tested a setup where the HN predicts the values of the batch normalization layers of the target Unet, in addition to the parameters of the last convolution layer. We name this **HyperSeg Conv + BN**. Second, we replaced the one-hot vector representation of each client, with a learned embedding vector. This embedding was trained jointly with the HN, and allows to capture the similarity structure across the set of clients. We name this variant **HyperSeg Conv + BN + Embedding**.

Training iterates between two phases: (1) *client stage*. Each client receives updated weights from the server, performs local training of the target network, and communicates network gradients to the server. (2) *server stage*, The server uses the gradients from all clients to update the HN and the global weights for the rest of the target network. See details in appendix C.

## 3 EXPERIMENTS AND RESULTS

We evaluated our approach using four different datasets as clients: Promise12 (Litjens et al., 2014), MSD-Prostate (Simpson et al., 2019), NCI ISBI 2013 Challenge, and PROSTATEx (Meyer et al., 2020). Figure 2, shows examples of scans form these datasets. Since each dataset was collected in a different settings, data distribution may vary across datasets. As we show below, a personalized model that can be adaptive to each data, is useful in this scenario. We followed the pre-processing and evaluation procedures used in Roth et al. (2021). The client target networks is a 3D-UNet architecture (Çiçek et al., 2016). The HN network is an MLP. See appendix B for the full details of the training process of the experiment.

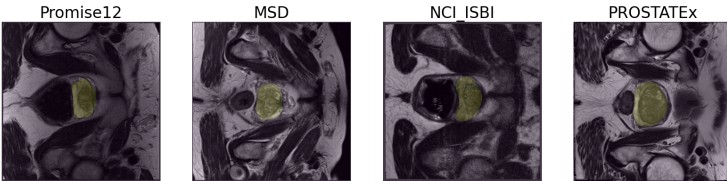

Figure 2: Prostate segmentation samples from 4 datasets. Yellow mask marks the ground truth.

Table 1 compares three variants of our approach described in Section 2, with two baselines. **(1) Local model:** Each client locally trains its own target network. **(2) Fed SGD** the fully-shared FL method from McMahan et al. (2017). All variants of our HyperSeg approach improve over baselnes, and HyperSeg+Conv+BN+Emb provides the strongest overall improvement. Additional experiments are given in Appendix C.

Table 1: Average Dice segmentation score, using two baselines and three variants of our approach.

| Avg. Dice [%] | Promise12 | MSD | NCI ISBI | PROSTATEx | Total Average |
|---|---|---|---|---|---|
| Local model | 12.13 | 50.53 | 24.5 | 61.83 | 37.20 |
| Fed SGD | 36.94 | 63.73 | 39.49 | 62.61 | 50.69 |
| HyperSeg Conv (ours) | 46.65 | **68.31** | 44.49 | 66.80 | 56.56 |
| HyperSeg Conv + BN (ours) | 45.07 | 67.18 | 48.27 | 66.53 | 56.77 |
| HyperSeg Conv + BN + Embedding (ours) | **50.03** | 67.93 | **48.72** | **68.07** | **58.69** |

URM STATEMENT

Here by I declare that I meet the following URM criterias: Age, Gender, Geographical criteria. It is also the first time for me to submit my work to an academic conference.

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

SUPPLEMENTAL MATERIAL

## A  DATA PROCESSING

We used data from four different sources: Promise12 (Litjens et al., 2014), MSD-Prostate (Simpson et al., 2019), NCI ISBI 2013[1] and PROSTATEx (Meyer et al., 2020). We split each data set to training set (70%), validation set (10%) and test set (20%). The resulting number of cases we use for each set can be found in Table 2. Each sample is being resampled to a resolution of 0.5mm $\times$ 0.5mm $\times$ 1.0mm and normalized by subtracting its mean intensity and dividing by its standard deviation, as in (Roth et al., 2021). While training, each sample was randomly cropped to size $160 \times 160 \times 32$, and an augmentation process was performed by randomly flip the sample (in all dimensions) as well as randomly scaling and shifting the intensities.

Table 2: Number of cases for each data set

| Cases | Promise12 | MSD | NCI ISBI | PROSTATEx |
|---|---|---|---|---|
| Training | 35 | 22 | 55 | 69 |
| Validation | 5 | 3 | 8 | 10 |
| Testing | 10 | 7 | 16 | 19 |
| Total | 50 | 32 | 79 | 98 |

## B  EXPERIMENT DETAILS

This section describe the details of the full training process for each experiment performed, and the settings differences between them. The computational resources allocated to experiments in the same table below are equal, to make the results comparable.

### B.1  SEGMENTATION ACCURACY

We first compare the two baselines and three variants of our approach, as described in Section 2. In our setup, all clients hold the same target network, 3D-UNet of size $(16, 32, 64)^2$, but differ in their training scheme. For the experiments that use a HN, we used an MLP with 20 fully-connected layers, each with 100 neurons.

Table 1 shows the results of five different training schemes: (1) local training of a 3D-UNet for each client, (2) federated learning of a 3D-UNet model using Federated SGD without HNs, (3) using HN to predict the weights of the last convolution layer only, and using Federated SGD for the rest of the parameters, (4) using HN to predict the batch normalization parameters alongside the weights of the last convolution layer, and (5) use HN to predict the above parameters while using an embedding vector of size 10 to represent the different clients.

For all training schemes we use a total of 22,250 training steps performed by all clients together, to optimize a binary cross-entropy loss function. In all of the experiments we use Adam optimizer for both optimizing the parameters of the HN (with learning rate of 0.03) and the target networks (with learning rate of 0.005).

As can be seen in table 1 above, considering the average Dice over all data sets (and for the majority of the data sets), each addition improves the performance of the model over the previous experiment.

### B.2  LARGER NETWORKS

We also tested another configuration of our approach. For all experiments we use a total of 44,500 training steps performed by all clients together, to optimize a binary cross-entropy loss function. In

---

[1]http://doi.org/10.7937/K9/TCIA.2015.zF0vlOPv

[2]this notation for the size of 3D-UNet means that the convolution blocks in the network include 16, 32 and 64 channels.

this configuration, all clients hold the same target network, 3D-UNet of size (32, 64, 128). For the training schemes that use a HN, we kept using an MLP with 20 fully-connected layers, each with 100 neurons.

Here we experimented three different training schemes: (1) Federated learning of a 3D-UNet model using Federated SGD without HNs, (2) using HN to predict the weights of the last convolution layer only, and using Federated SGD for the rest of the parameters, and (3) using HN to predict the batch normalization parameters alongside the weights of the last convolution layer.

Table 3 shows the performance of the different training schemes for this configuration. As can be seen, the addition of the HN and its ability to predict the batch normalization parameters improves the average performance over all datasets and the performance for the majority of the datasets.

Table 3: Average Dice segmentation score

| Avg. Dice [%] | Promise12 | MSD | NCI ISBI | PROSTATEx | Total Average |
|---|---|---|---|---|---|
| Fed SGD | 54.44 | 70.13 | 51.18 | 70.88 | 61.66 |
| HyperSeg Conv (ours) | **57.34** | 68.51 | 53.33 | 70.22 | 61.66 |
| HyperSeg Conv + BN (ours) | 55.27 | **74.50** | **55.64** | **71.34** | **64.19** |

### B.3 CLIENT-SPECIFIC NOISE.

We also tested our approach using another variation of the four datasets. we added a unique noise for each dataset, typical to MRI scans, resulting in a more defined separation between the data distributions among clients. Full description of the noise can be found in Appendix D.

Here we used a total of 44,500 training steps, and all clients hold the same target network: 3D-UNet of size (16, 32, 64, 128). Again, for the training schemes that use a Hypernetwork, we used an MLP with 20 fully-connected layers, each with 100 neurons.

Table 4 shows the segmentation quality when we added client-specific noise for different training schemes we used. Here, incorporating the HN improves the average performance over all datasets and the performance for the majority of the datasets with respect to the baseline Federated SGD training.

Table 4: Average Dice segmentation score for client-specific noise

| Avg. Dice [%] | Promise12 | MSD | NCI ISBI | PROSTATEx | Total Average |
|---|---|---|---|---|---|
| Fed SGD | 52.57 | 72.48 | **55.74** | 67.99 | 62.20 |
| HyperSeg Conv (ours) | **58.47** | **73.72** | 53.30 | **70.92** | **64.11** |
| HyperSeg Conv + BN (ours) | 54.03 | 72.12 | 55.61 | 69.92 | 62.92 |

## C  CLIENT SERVER COMMUNICATION WORKFLOW

**Client side.**   The client makes a request to receive updated model weights from the server. Then, the server uses the embedding vector of that client as an input to the Hypernetwork, predicts the *personalized weights* of the target network for that client, and sends them to the client. The server also sends the global weights to the client. To improve stability and reduce communication cost (Shamsian et al., 2021), the client performs several training steps for its local target network, using its local data. The client then sends the gradients for all weights, personalised and global, to the server.

**Server side.**   Receiving the updated gradients from all clients, triggers two processes: (1) First, the server trains the Hypernetwork and tunes the embedding vectors using the received gradients, as in Shamsian et al. (2021). (2) The server updates the global weights using a (non personalized) FL algorithm. In our experiments, we use Federated SGD (McMahan et al., 2017) as our FL technique for learning the global weights. As a future research, one can use other federated learning methods, like SpitNN by Vepakomma et al. (2018).

## D  CLIENT-SPECIFIC NOISE DESCRIPTION

We further evaluated our method when adding unique noise for each dataset. The noise types we use are implemented in the MONAI[3] framework.

We used two types of noise: (1) *Gibbs Noise*, that appears as ripples or oscillations near sharp edges in the MRI image, was added in two different magnitudes to samples from Promise12 dataset (intensity of 0.2) and from PROSTATEx dataset (intensity of 0.5). (2) *Spike noise in k-space*, which may results as bright spots in the resulting MRI image, was added to samples from the NCI ISBI dataset (intensity of 13). Since our goal is to make a clear separation between the data distributions of the clients, we added noise to 3 out of 4 datasets and did not add any noise to the MSD-Prostate dataset.

Figure 3: Prostate segmentation samples with and without noise addition

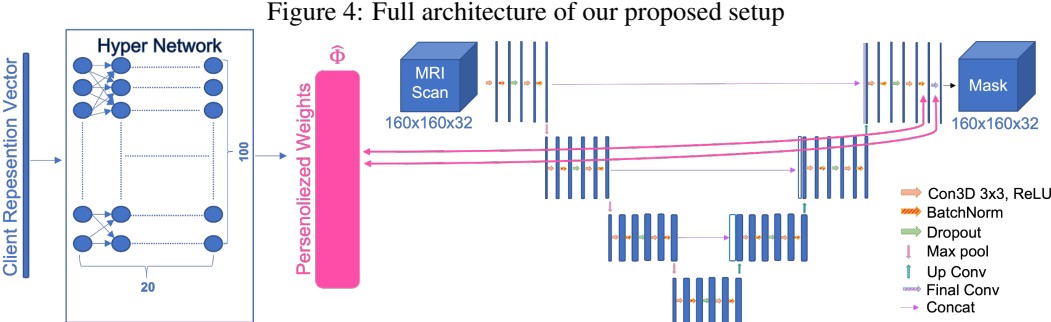

## E  FULL ARCHITECTURE

Figure 4: Full architecture of our proposed setup

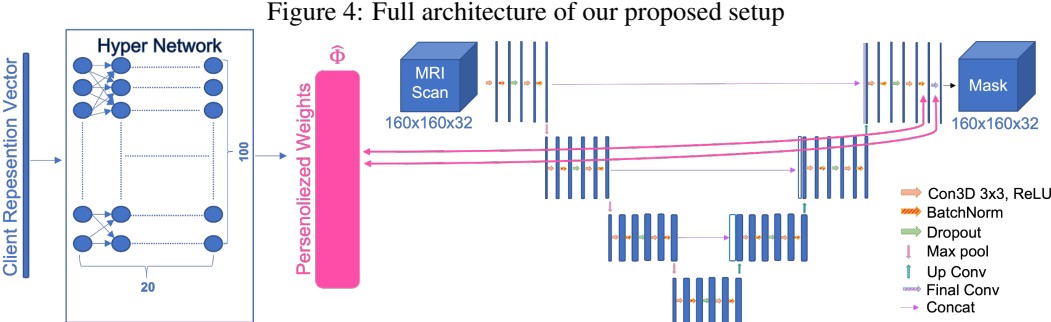

---

[3]https://monai.io/

