# OpenReview forum: "Personalized Federated Learning for Medical Segmentation using Hypernetworks"
_ICLR.cc/2023/TinyPapers — Submitted to Tiny Papers @ ICLR 2023_

### Official Review · Reviewer_AtVj · 2023-03-27

**Confidence:** 3

**Summary Of Contributions:**

The paper presents a preliminary case study that applies Hypernetworks and Personalized Federated Learning towards solving segmentation task in medical images. The main contribution of the work is to verify empirically that such setting, which has been tested successfully with easier classification tasks, can be extended to more complicated segmentation task.

**Rating:**

Great Start (GS): a submission which meets some of the reviewing criteria but has room for improvement

**Strengths And Weaknesses:**

Strengths:
1. The paper is well-written, explains the problems and the settings well.
2. Though straightforward in terms of ideas and research directions, the presented study may be more valuable to the applied medical tools.

Weaknesses:
1. The idea is very simple, given that the HN+PFL setting has been tested as helpful for the classification problems, testing the exact same setting just with a segmentation task seems a bit lack of novelty.

**Suggested Changes:**

The paper provides a great start for doing research in applying federated learning within medical image analysis, I think that the work presented in this paper serves more like a baseline model (existing setting + new task) for authors to developing a new, better setting for this new task. There isn't much to change slightly for the current version, but I encourage the authors to keep pushing for new results.

---

> ### Author Response · Authors · 2023-04-20
> **Response**
>
> We thank the reviewer for taking the time to thoroughly review our paper.
>
> Following is our comment regarding the mentioned weakness of our research:
>
> > The idea is very simple given that HN+PFL has been tested for classification.
>
> The challenges with applying PFL-HN to medical segmentation are two-fold. First, segmentation models are very large and it is not clear what parts of these models should be generated by the HN, and which should be globally shared. Second, this paper addresses the problem of covariate shift across clients, while the approach of Shamsian et al looked into PFL-HN for label-shift across clients, while the distribution of the p(x|y) is preserved across clients.
> We explored several methods to handle this challenge. First, we pinpointed which subset of the weights should be personalized by the HN. Then we also proposed a procedure based on a global FL method (Fed-SGD) to train the shared weights of the model.

---

### Official Review · Reviewer_8Amc · 2023-04-01

**Confidence:** 4

**Summary Of Contributions:**

This paper proposes a use-case for hypernetworks in personalized federated learning for segmentation of prostate MRI scans

**Rating:**

Great Start (GS): a submission which meets some of the reviewing criteria but has room for improvement

**Strengths And Weaknesses:**

I would first like to commend the author(s) for this work especially given that this is a first-time submission to an academic conference. This is a well-structured, concise paper that is supported nicely with suitable experiments.

Clarity :
- findings are communicated clearly and effectively.
- relevant literature has been included though this section can be improved.

Reproducibility: Code not provided. Publicly available datasets have been used.

Follows basic requirements: Yes

Correctness: The paper provides the technical details of the implementation. It is a good start and can be improved further by addressing these comments:

- the claim that "We describe the first HN-based PFL for segmentation of medical images" may not hold true (though I am happy to be proven wrong) - please refer to the references given in the next section. Especially for the first reference (Wang and Jin et al.), please highlight the differences between their approach and the framework proposed in this paper.
- In the HyperSeg Conv setup, was also the 'HyperSeg Conv + Embeddings' setup evaluated ? If yes, please include them in the results table.
- If possible, in the evaluation metrics, please include IoU (for regions) and ASSD scores (for boundaries) - again, refer to the 1st reference in the next section.
- For the training, please mention the hardware details and the training/inference times.
- Does the learning rate stay constant or does it reduce as training progresses ?
- Have the authors explored more recent approaches such as Split-learning that could be used with HN instead of Federated SGD ?


**Suggested Changes:**

1. Please refer to the following references and add them to the paper where necessary:
- regd segmentation (minus HN). This submission (to tiny papers) focuses on using PFL+HN settings that have been used for classification and extending them to a segmentation task. This paper would be extremely helpful to refer to the opposite - using the settings that work for prostate segmentation + pfl and combining them with HN: https://www.ecva.net/papers/eccv_2022/papers_ECCV/papers/136810449.pdf
- split learning: https://aiforsocialgood.github.io/iclr2019/accepted/track1/pdfs/31_aisg_iclr2019.pdf
- FL with HN but classification (models used): https://www.nature.com/articles/s41598-023-28974-6
- FL with HN for reconstruction (requires larger models): https://arxiv.org/pdf/2206.03709.pdf

2. We thank the authors for the kind words in the last paragraph in their URM statement. As a suggestion, I would recommend having this paragraph as a footnote if there are any space constraints in case any of the review comments are incorporated.

3. Please mention the hardware details and training times. Also, please address some of the points mentioned in previous section under the 'correctness' sub-section.

If the comments are addressed or clarified reasonably, I would be happy to consider upgrading the assigned score.

---

> ### Author Response · Authors · 2023-04-20
> **Response**
>
> We thank the reviewer for taking the time to provide such detailed comments and suggestions on our paper.
>
> Following are our comments:
>
> *Comments:*
> > Please highlight the differences between the approach of (Wang and Jin et al.) and the framework proposed in this paper.
>
> Both methods address the problem of segmentation in PFL.
>
> A major difference is that our approach uses Hypernetworks for learning to share parameters across clients, while the paper by Wang et al uses a different method that does not involve HNs. Specifically, their approach is based on learning local attention over features.
>
> Indeed, our approach is not the first to do PFL for segmentation but is the first to use HNs for this task.
>
> > In the HyperSeg Conv setup, was also the 'HyperSeg Conv + Embeddings' setup evaluated ? If yes, please include them in the results table.
>
> Following this comment, we will add these results to the camera-ready version.
> Specifically, for the “Larger Networks” setup, the average Dice scores are as follows: Promise12: 61.22%, MSD: 75.68%, NCI ISBI: 60.16%, PROSTATEx: 71.8%, Total Average: 66.22%.  These results further improve the results presented in the submission (Table 3).
>
> > For the training, please mention the hardware details and the training/inference times.
>
> We used a single V100-SXM2-32GB NVIDIA GPU. The entire workflow lasted 36 hours. This included training and repeated evaluations over the validation set every 10 epochs and evaluation of the entire test set every 50 epochs.
>
> > Does the learning rate stay constant or does it reduce as training progresses?
>
> We did not use scheduling of the learning rate. Note that we did use an Adam optimizer, which adjusts the gradient step size based on past gradients.
>
> > Have the authors explored more recent approaches such as Split-learning that could be used with HN instead of Federated SGD?
>
> Thank you for the great idea, we will look into this approach.
>
> *Suggested Changes:*
> > Please refer to the following references and add them to the paper where necessary.
>
> Thank you for the very useful references. We will add a new section to the supplement information, covering additional relevant literature.

---

### Meta-Review · Area_Chair_b5N4 · 2023-04-04

**Recommendation:** Invite to present
**Confidence:** 4

**Metareview:**

The paper explores the use of hypernetworks and personalized federated learning for medical imaging. It introduces the premise and experimental setup clearly. The level of detail provided is admirable making this work reproducible.

**Summary:**

The paper explores the use of hypernetworks and personalized federated learning for medical imaging. The authors demonstrate that using a hypernetwork to personalize a single convolution layer and the batch-norm layer outperforms local and FL baselines.

**Reason For Not Giving A Higher Recommendation:**

One reviewer mentioned lack of novelty as a concern but in the spirit of TinyPapers this is not a priority. However, if the authors can address the concerns mentioned by the other reviewer, this will make for a great presentation.

**Reason For Not Giving A Lower Recommendation:**

The paper is a good submission to this venue.

---

### Decision · Program_Chairs · 2023-04-07

Invite to present